# In Situ Radioactivity Maps and Trace Metal Concentrations in Beach Sands of a Mining Coastal Area at North Aegean, Greece

Christos Tsabaris [1],*, Dionisis L. Patiris [1], Rosalinda Adams [2], Julian Castillo [2], Maria F. Henriquez [2], Caroline Hurtado [2], Lesley Munoz [2], Leonidas Kalpaxis [2], Mariana Verri [2], Stylianos Alexakis [1], Filothei K. Pappa [1,3] and Angelos Lampousis [2]

1. Hellenic Center for Marine Research, Institute of Oceanography, 46.7 Km Athens-Sounio Ave, 190 13 Anavyssos, Greece; dpatiris@hcmr.gr (D.L.P.); salexakis@hcmr.gr (S.A.); fkpappa@gmail.com (F.K.P.)
2. Department of Earth and Atmospheric Sciences, City College of New York, City University of New York, New York, NY 10031, USA; adams004@citymail.cuny.edu (R.A.); jcastil011@citymail.cuny.edu (J.C.); mhenriq004@citymail.cuny.edu (M.F.H.); churtad000@citymail.cuny.edu (C.H.); lmunoz000@citymail.cuny.edu (L.M.); lkalpax000@citymail.cuny.edu (L.K.); maryvcalix@gmail.com (M.V.); alampousis@ccny.cuny.edu (A.L.)
3. Department of Marine Sciences, University of the Aegean, 811 00 Mytilene, Greece
* Correspondence: tsabaris@hcmr.gr; Tel.: +30-2291076410

**Abstract:** In recent years, the environmental effects of both active and legacy mining activity have motivated many research groups worldwide through the use of a variety of methods that have been conducted among diverse environments. In this study, we measured radionuclide concentrations at two coastline locations of the Northern Aegean Sea: Stratoni and Ierissos. We deployed KATERINA II, an in situ gamma ray spectrometer. Our results indicate that the activity concentration for $^{238}$U progenies ($^{214}$Bi), $^{232}$Th progenies ($^{208}$Tl and $^{228}$Ac) and $^{40}$K vary by up to $(33 \pm 4)$ Bq kg$^{-1}$, $(19 \pm 3)$ Bq kg$^{-1}$, and $(420 \pm 30)$ Bq kg$^{-1}$, respectively. The activity concentration of the $^{137}$Cs in Stratoni and Ierissos beach sands were $(8.1 \pm 2.2)$ and $(3.9 \pm 1.2)$ Bq kg$^{-1}$, respectively. Lab-based measurements were also collected prior to the in situ data collection for the determination of radionuclide and metal concentrations. The lab-based data were found to be $(800 \pm 40)$ µg g$^{-1}$ and $(12 \pm 1)$ µg g$^{-1}$, for As, $(1200 \pm 60)$ µg g$^{-1}$ and $(33.3 \pm 0.3)$ µg g$^{-1}$ for Pb, $(100 \pm 6)$ µg g$^{-1}$ and $(6.0 \pm 0.3)$ µg g$^{-1}$ for Cu and $(2000 \pm 60)$ µg g$^{-1}$ and $(8.0 \pm 0.4)$ µg g$^{-1}$ for Zn, respectively. We used the R language and environment for statistical computing to produce radiological maps of the subject beach sands. We used the Enrichment Factor (EF) to estimate assessment indices for the target area and compared them to internationally recommended values. The in situ maps will be discussed since the beach area of the load-out pier area of Stratoni was undergoing the first phase of active remediation. We conclude that the temporal aspect of this dataset can be of significant reference value against future comparative studies after the remediation of the Stratoni beach with potentially denser spatial and temporal data coverage.

**Keywords:** beach sand; in situ instrumentation; natural radionuclides; radioecology; coastal mining

## 1. Introduction

Naturally occurring radioactive materials (NORMs) have existed since the formation of the Earth's crust, and are found in different amounts in the environment [1]. NORMs are typically long-lived radionuclides, and their intensity is the result of natural sources, such as the $^{238}$U and $^{232}$Th series and their progenies, in addition to $^{40}$K [2]. Another component of NORMs is produced by the interaction of cosmogenic radiation with the atmosphere and its deposition onto the planet surface. The radioactivity levels of NORMs exhibit spatial and temporal gradients due to the geochemical composition and geophysical conditions of the Earth's crust affecting sediments, beach sands, and rocks. In addition to the naturally occurring radioactivity, $^{137}$Cs is widely used as a radiotracer in various oceanographic studies and processes that take place in the coastal environment.

NORMs are typically reported in the literature in support of efforts to study aquatic systems [3,4]. Recent studies have focused on the determination of the activity concentration of natural radionuclides in sediments to assess sediment transfer times in river systems, as well as to reconstruct historic river flooding events using thorium radiotracers [5]. Other studies have focused on the use of natural radionuclides in sediments as tracers for the identification of transport pathways though sediments in coastal areas [6–8]. In recent years, the reconstruction of past events [9–11] have been performed, analyzing core sediments in terms of sedimentation and accumulation rates to assess the temporal evolution of the environmental status and how is related with global environmental priorities such as climate change [12] and marine pollution studies [10,11]. $^{137}$Cs is an artificial radionuclide with a half-life of 30.1 years. Its presence in the environment is of great concern due to its effect on human life [13]. It is produced during the nuclear fission process, and has a relatively high fission yield. $^{137}$Cs emits at an energy of 661.66 keV gamma rays, which is an easily detectable photopeak when using the gamma-ray spectroscopy detection method with low, medium, and high resolution detectors [13]. The presence of $^{137}$Cs in the environment may be traceable to nuclear weapons testing during the period of 1955-1963 [13], a period of deposition from 1971 to 1974 [14], and the nuclear power plant disasters of Chernobyl (1986) and Fukushima (2011) [15]. Global $^{137}$Cs fallout concentrations have been used in studies of soil erosion, sediment accumulation rates, and sediment dynamics issues that take place in the coastal zone [14]. The activity concentration of $^{137}$Cs in a worldwide basis is site-specific, with the northern hemisphere indicating a greater amount than the southern. Beaches are affected from fallout processes and/or from various incidents that took place in the nuclear industry in recent decades and play crucial role in protecting humans in terms of radioactive contamination. $^{137}$Cs is considered to be one of the most critical radionuclides for assessing the health of the environment.

Active mining activities located in Stratoni include the mines of Madem Lakkos, Mavres Petres, Olympias [11], the metal floatation plant in the coastal part of Stratoni town, the loading docks in Stratoni's port, and solid waste byproducts [16]. The mine tailings were dispersed directly into the nearshore of Stratoni's marine environment until 1983 [16]. The mining activity in Stratoni has a long history dating from 600 BC. Mining and metallurgical activities peaked in the 1970s. The environmental monitoring of the region was primarily administered by an exploration company [16], and to a lesser extent by research groups [9], emphasizing the heavy metal concentrations within sediments situated in proximity of the shoreline. As far as we are aware, data concerning radioactivity measurements in beach sands at the aforementioned mining area are scarce. Environmental remediation efforts in the beach sand area of Stratoni were initiated shortly prior to the time of our own data collection activities.

## 2. Material and Methods

### 2.1. Study Area and Fieldwork

In this study, we contrasted the activity concentrations of natural and artificial radionuclides between two neighboring coastlines of Northern Greece. The first one is Stratoni, which is characterized by legacy and active mining activities. The second site is Ierissos, which is primarily residential, with no prior mining or other industrial history. The evaluation of the radiation dose and its impact on humans is based on estimating the concentration of both natural and artificial radionuclides, as well as the enrichment factors of the metal concentration. In our study, the radiological assessment is performed using the geo-referenced in situ detection system KATERINA II, which enables the rapid mapping method of the key radionuclides to produce pre-remediation data sets and corresponding maps. Additionally, the metal concentrations of key mining pollutants were measured in the laboratory. As concerns the sampling program, a rapid mapping method of the beach sand was organized to characterize the specific sites. Previous lab-based measured data of the activity concentration of radionuclides as well as of trace metals were also reported to better identify the advantages of the in situ method, as well as to produce a set

of measurements and an initial assessment in a pre-remediated beach sand area that was significantly affected by mining activities during the last century.

The study areas (the Stratoni and Ierissos beaches) are situated in the Ierissos Gulf in the Chalkidiki Peninsula in northern Greece (Figure 1). A description of the study area's characteristics is included in the details elsewhere [9,11]. Briefly, the city of Stratoni is located at the north part of the Ierissos Gulf, where a polymetallic mine is located. In this mine, the primary minerals (pyrite ($FeS_2$), sphalerite/zinc blende (ZnS) and galena (PbS)), and the secondary ones (arsenopyrite (FeAsS), rhodochrosite ($MnCO_3$), chalcopyrite ($CuFeS_2$), bornite ($Cu_5FeS_4$) and magnetite) [16] were exploited from ancient years until the present day. At the Ierissos gulf, rivers and streams also discharge mining residuals due to the floatation plant operation. The city of Stratoni is also located at the load out pier area of the mine. On the other hand, Ierissos is situated in the southern part of Stratoni, and it mainly hosts tourist activities. A detailed description of the study areas can be found in the literature [9,11].

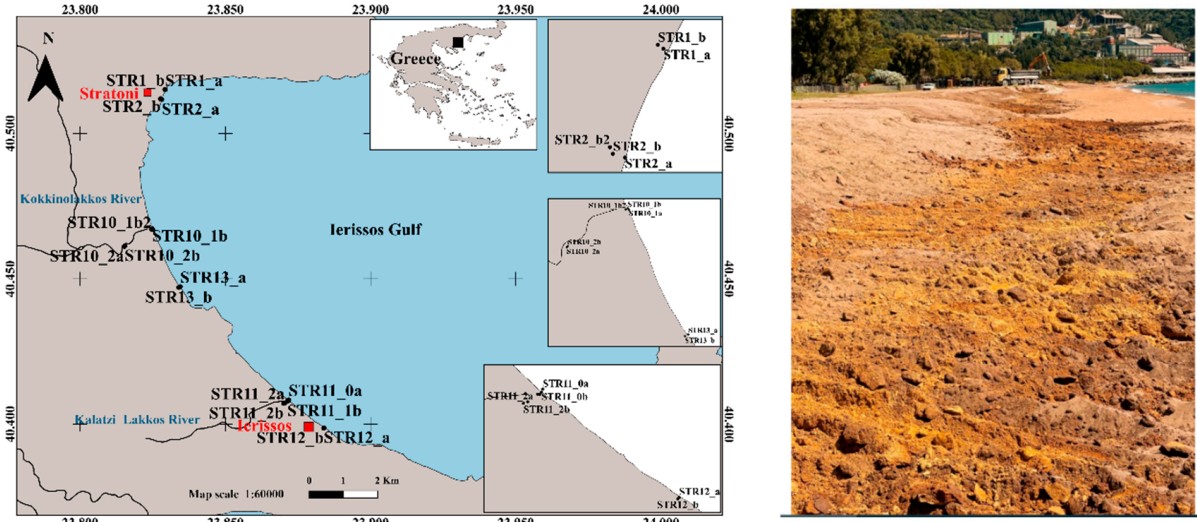

**Figure 1.** The map of the study area at the Stratoni and Ierissos sites (**left**), and the picture of the remediation activity (**right**). The lab-based measurements were performed previously prior to the in situ measurements, and all sampling points are depicted in the map.

The beach sand that is located in the site of Stratoni is undergoing a remediation process due to the past mining activities. In Figure 1, a typical photo (right) depicts the various byproducts that were present before remediations activities, due to the past mining activities that produced oxidation of metals together with copper, sulfides, and oxides [10,11,16].

### 2.2. KATERINA II Detection System

The development of gamma ray detection systems capable for in situ and on-site measurements in the marine environment is of great importance due to the great variety of their applications. In that regard, the detection system KATERINA II [17,18] was developed to support studies related to oceanographic processes using radionuclides as tracers [19–21]. The KATERINA II detection system consists of a 3″ × 3″ NaI scintillator with negligible intrinsic activity, connected with a photomultiplier tube, a preamplifier combined with appropriate units for signal amplification, data acquisition, and storage. The NaI crystal represents a low cost solution that is compatible to the selected method of quantification. The output of the whole detection system is connected to a computer via three commonly used communication protocols: Serial RS232, USB, and Ethernet. A volatile memory and a microcontroller are incorporated in the system in order to be independent of any computer connection. The data logger consists of a compact stand-alone digital Multi-Channel Analyzer with a fully controlled microprocessor to perform data acquisition using digital

signal processing algorithms. The electronic modules require low power consumption (~1.0 W). The KATERINA II system is designed to be easily integrated in any fixed station, floating and/or mobile platform, allowing it to operate in a stand-alone as well or a near real-time mode.

The KATERINA II detection system has been upgraded to rapidly perform gamma-ray spectrometric measurements in a spatial manner, integrating geo-referenced information. The system operates using a low-resolution crystal and appropriate electronics for geospatial mapping in coastal areas. The system also integrates a method to offer the activity concentrations of all detected gamma-ray emitters in absolute units, combining simulation runs for system efficiency. The survey transects of the spectrometer were collected by mounting the in situ system as a backpack in order to acquire the gamma ray spectra. The radiometric maps on the beach sands were produced after the experimental work. The system was deployed across the above-mentioned coastal environments. The detection system was programmed via the mini-computer that integrates the GPS with the detector itself before the experiment to provide sequential time-series measurements. The power was supplied by a mini battery that was loaded in the user's back-pack.

*2.3. Field Work and Lab-Based Techniques*

Prior to the in situ NORM measurements, point data were collected to determine the metal (minor and major) and activity concentrations. Indicative samples were collected from the near-surface sediment (0–5 cm) from the beach and the beach-sea interface (some meters away from the berm area of the beach sites). Sediments from the two main rivers of the Ierissos Gulf were collected to better understand the influence of the surrounding rivers. The adopted lab-based methodologies (the X-ray fluorescence and gamma-ray spectrometry laboratories of the Hellenic Centre for Marine Research) were applied for the trace and major metal concentrations and the NORM concentrations. These methodologies afford high levels of accuracy, but require extensive sample preparation and treatment, which results in long acquisition measurement times compared to other rapid determination applications. The samples were treated according to the literature [9–11]. The trace metal concentrations constituted the reference for determining the enrichment factor for the subject areas. Ierissos beach was selected as a pristine area, since it exhibits the lowest trace metal concentrations compared to continental crust shale [22]. Al (aluminum) was used as a normalizing element due to the fact that it was characterized by the minimum coefficient of variation compared to the other studied major elements.

The activity concentration of radionuclides was measured using a calibrated 50% (nominal relative efficiency) broad energy HPGe n-type coaxial detector (ORTEC) with an ultrathin carbon-fiber window and a resolution of 1.85 keV at 1.33 MeV. A lead shield surrounded the detector to reduce the ambient gamma ray background. The detection system has been calibrated for the measured geometry using appropriate reference sources ($^{152}$Eu/$^{154}$Eu as well as $^{238}$U/$^{235}$U) enriched in similar material in terms of density, covering the energy interval of 45–2000 keV. Detailed information concerning the use of the detector in extended geometries is described elsewhere [23].

*2.4. Mapping Method*

The mapping of the total counting rate obtained by the mobile in situ gamma-ray spectrometry method was carried out for the rapid characterization of the area in order to assess the dose for humans, in addition to the environmental status. For the mapping, the Inverse Distance Weighted (IDW) method was selected, as it is a widely used deterministic method. The main underlying concept of the method is "Tobler's first law of geography" which in this specific case applies to the influence of neighbour points to the TCR value of a selected point. The estimation of the TCR value of a selected point ($TCR_p$) is more

influenced by its neighbour points than by its far points. The method was applied based on the following mathematical formula:

$$TCR_p = \frac{\sum_{i=1}^{n}\left(\frac{TCR_i}{d_i^p}\right)}{\sum_{i=1}^{n}\left(\frac{1}{d_i^p}\right)}$$

where $n$ is the amount of the obtained data, and $d_i^p$ is the distance between a selected point among them. The power ($p$) reflects the strength of the influence of the neighbour data, and how the influence decays with the distance. A value between two and three has been demonstrated to be appropriate in order to reproduce the distance dependency in similar surveys [24]. The production of the maps and the calculations were resulted from the use of a code written in the R language [25], which provides a wide variety of statistical and graphical techniques under the terms of the Free Software Foundation's GNU (General Public License).

*2.5. Dose Rate Assessment*

The dose rate is assessed in terms of natural radionuclide materials (NORMs), since the activity concentration of $^{137}$Cs is currently (after one half-life of the isotope) at a background level. The dose rates were calculated using the following equation according to the literature [26]:

$$\dot{D} = 0.042 \cdot A(^{40}K) + 0.429 \cdot A(^{238}U) + 0.666 \cdot A(^{232}Th) \tag{1}$$

where $A(^{40}K)$, $A(^{238}U)$ and $A(^{232}Th)$ are the activity concentrations of $^{40}K$, $^{238}U$ and $^{232}Th$ in Bq kg$^{-1}$, respectively, and $D$ is dose rate in nGy h$^{-1}$. The annual dose is simply calculated for the days that humans are exposed to the studied ecosystem.

## 3. Results

The Gamma-ray surveys were performed by integrating/mounting the geo-referenced low resolution gamma-ray spectrometer in a backpack, enabling the mobile method to provide the gamma-ray spectra for each time lag. The natural and anthropogenic radionuclide concentrations have been estimated using regions of interest and simulated efficiencies. The results are related with the activity concentration of NORMs and $^{137}$Cs enabling the in situ method and the use of the KATERINA II system. The measured data will be categorized for the two study areas.

*3.1. Stratoni Port*

The map of the total counting rate (TCR) using data interpolation methods is provided for the Stratoni beach area in Figure 2. The total counting rate varied from 150 to 510 counts per second in the entire area. A total of 345 spectra were acquired during the experiment, and the live time of all measurements was 6900 s. In Figure 2, the interpolation data are classified with different colors, while the experimental data are depicted with colored circles in order to show the experimental data via the interpolation process. The areas exhibit the highest values in an area where a dynamic system discharges material at the west part of the beach. On either side of the berm zone, we observed various unearthed foreign sediment types, which we identified as the likely by-products of mining activities. Environmental remediation activities (i.e., excavation and top-soil removal) were underway throughout our data collection program. The total spectrum as acquired during the whole campaign is depicted in Figure 3, and is given automatically from the developed software using the R code [25]. We consider the total spectrum in order to determine the activity concentrations of NORMs and $^{137}$Cs, since the statistics are adequate for data analysis in terms of average values in the entire area. At the bottom-left area of the spectrum (Figure 3), a zoom process was performed to demonstrate the energy peaks at low energies

(till 800 keV). In qualitative terms, the spectrum mainly contains mainly the typical NORMs, such as the progenies of $^{238}$U ($^{214}$Pb and $^{214}$BI) as well as the progenies of $^{232}$Th ($^{228}$Ac and $^{208}$Tl). The total spectrum was analyzed for $^{137}$Cs using the SPECTRW software package [27]. At the high gamma-ray energies of the spectrum (the right part of Figure 3), possible sum peaks appear (e.g., due to the sum of 2614 keV of $^{208}$Tl and 583 keV of $^{208}$Tl, which are not considered for further analysis). The energy photopeak around 600 keV is the result of the contribution of three radionuclides: $^{214}$Bi at 609 keV, $^{208}$Tl at 583 keV, and $^{137}$Cs at 661 keV. The photopeak efficiency is calculated using the EGS4nrc simulation code [28] without taking into consideration the human body (where the detection system was mounted during the experimental work) in the simulation runs.

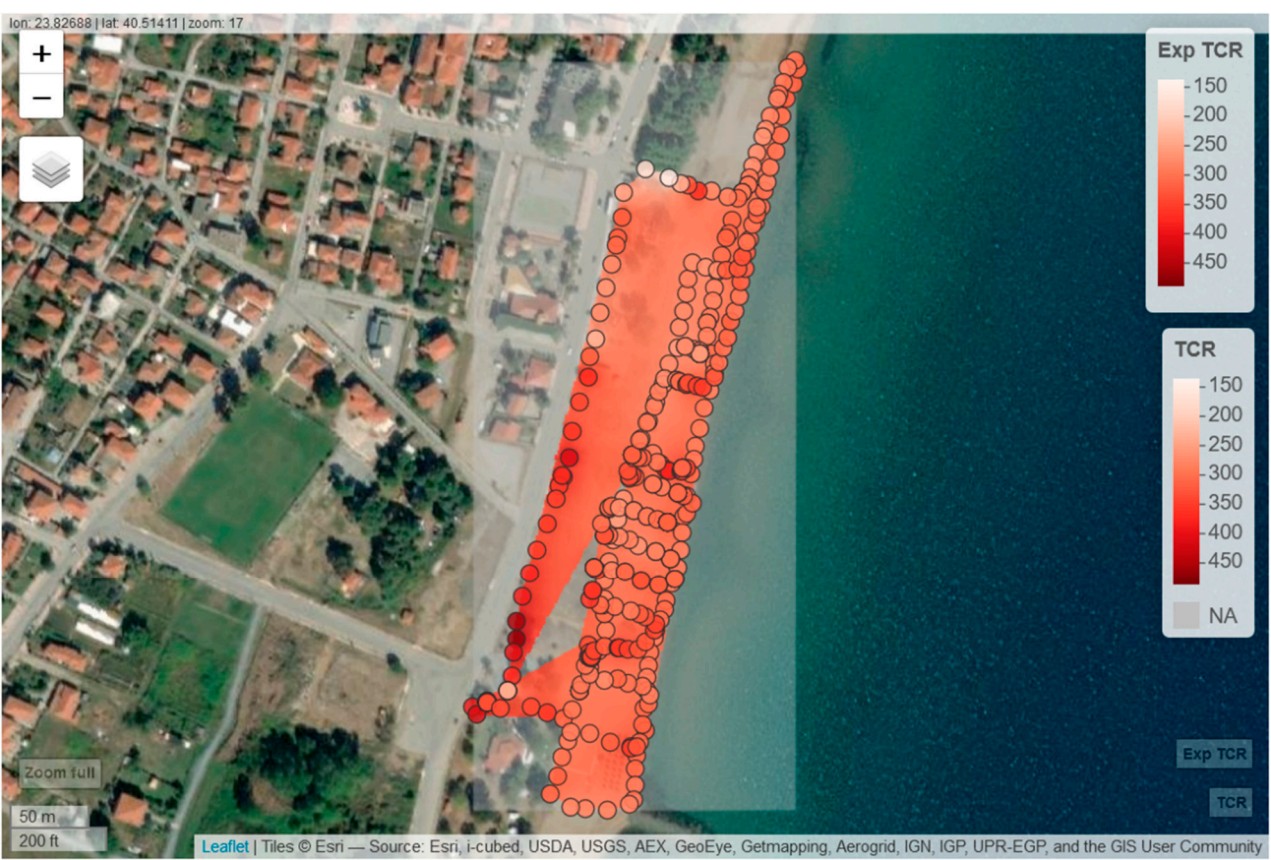

**Figure 2.** The radioactivity distribution of the total counting rate (TCR) using interpolation methods (Stratoni area). The experimentally derived TCR are depicted with circles.

As concerns the laboratory analysis and the results of activity concentrations, the samples were collected in the beach sand and the beach-sea interface (the berm area as given in Figure 1). The activity concentrations in the Stratoni area for $^{238}$U progenies ($^{214}$Pb, $^{214}$Bi, $^{226}$Ra) and $^{232}$Th progenies ($^{208}$Tl and $^{228}$Ac) were approximately (50 ± 4) Bq kg$^{-1}$ and (25 ± 2) Bq kg$^{-1}$, respectively. The $^{40}$K concentrations ranged from (340 ± 40) Bq kg$^{-1}$ to (730 ± 70) Bq kg$^{-1}$, respectively. The activity concentration of $^{137}$Cs exhibited values of (0.8 ± 0.2) Bq kg$^{-1}$. With regard to the trace metal concentrations in the Stratoni beach area, their uncertainties were around 5%, and the measured values of As (arsenic), Pb (lead), Zn (zinc), and Cu (copper) ranged from 530 to 4300 μg g$^{-1}$, 600 to 1800 μg g$^{-1}$, 1300 to 3900 μg g$^{-1}$, and 80 to 300 μg·g$^{-1}$, respectively. The major element concentration uncertainties were approximately 5%, and the measured values of Al (aluminum), Fe (iron), S (sulphur), and Mn (Manganese) ranged from 27 to 48 × 10$^3$·μg g$^{-1}$, 38 to 200 × 10$^3$·μg g$^{-1}$, 15 to 26 × 10$^3$·μg g$^{-1}$, and 8 to 26 × 10$^3$·μg g$^{-1}$, respectively. All measured data of the trace metals in the area of Stratoni are listed in Table 1.

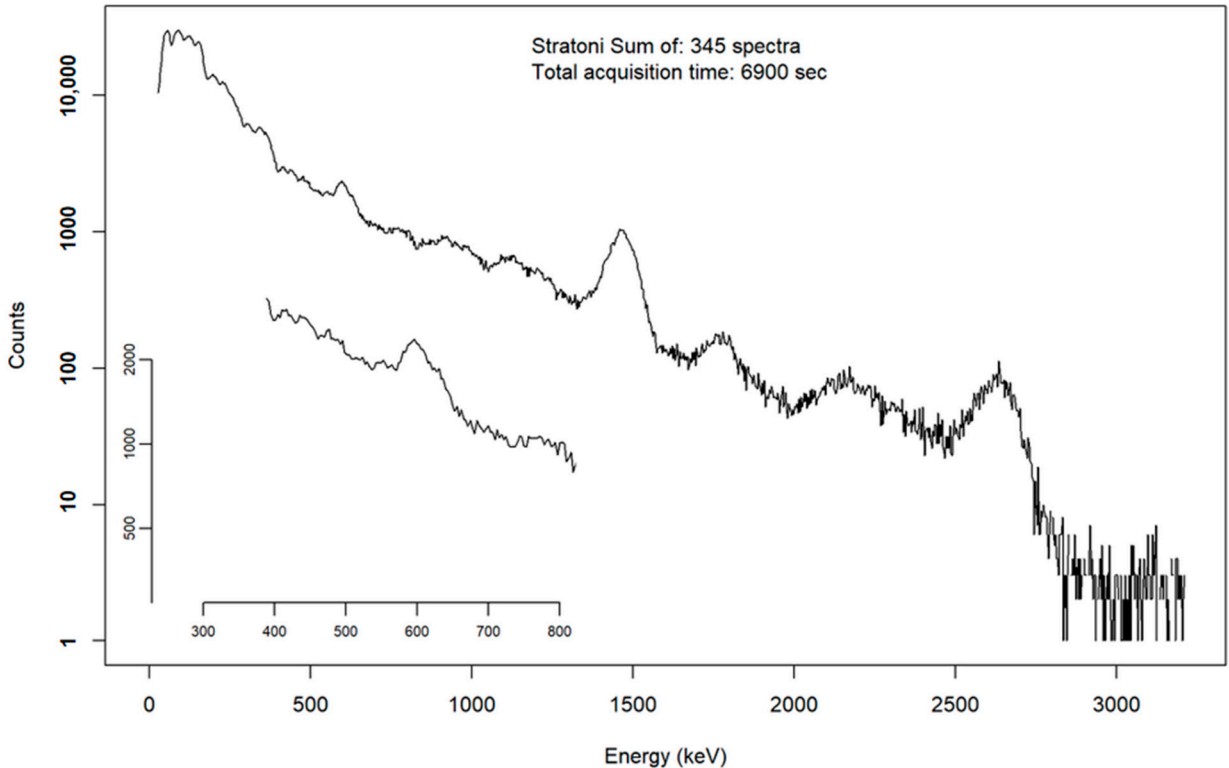

**Figure 3.** The acquired spectrum in the beach of Stratoni using the KATERINA II system. At the middle of the spectrum (1461 keV), the intense contribution of $^{40}$K is present. At 1760 and 2614 keV, the contributions of $^{214}$Bi and $^{208}$Tl, are observed for further analysis, respectively.

**Table 1.** The trace metal (As, Cu, Pb, Zn), Mn concentrations, and the determined enrichment factors. The relative uncertainty is 5%.

| Station | Trace Metal Concentrations | | | | | Enrichment Factors | | | | |
|---|---|---|---|---|---|---|---|---|---|---|
| | **As** | **Cu** | **Pb** | **Zn** | **Mn** | **As** | **Cu** | **Pb** | **Zn** | **Mn** |
| | $\mu g \cdot g^{-1}$ | $\mu g \cdot g^{-1}$ | $\mu g \cdot g^{-1}$ | $\mu g \cdot g^{-1}$ | $\mu g \cdot g^{-1}$ | | | | | |
| *STR1_a* | 1500 | 140 | 1200 | 3900 | 27,000 | 197 | 37 | 58 | 736 | 482 |
| *STR1_b* | 800 | 80 | 1100 | 2000 | 9200 | 79 | 17 | 41 | 282 | 127 |
| *STR2_a* | 530 | 110 | 600 | 1300 | 8000 | 53 | 23 | 22 | 187 | 112 |
| *STR2_b* | 570 | 120 | 830 | 1900 | 17,000 | 63 | 27 | 33 | 308 | 266 |
| *STR2_b2* | 4300 | 300 | 1750 | 3800 | 8000 | 750 | 109 | 112 | 987 | 206 |
| *STR10_1a* | 200 | 30 | 400 | 570 | 3500 | 22 | 6 | 14 | 83 | 48 |
| *STR10_1b* | 300 | 50 | 650 | 890 | 5800 | 29 | 9 | 22 | 120 | 75 |
| *STR10_1b2* | 980 | 90 | 2600 | 1100 | 19,000 | 106 | 21 | 102 | 182 | 293 |
| *STR10_2a* | 260 | 80 | 510 | 950 | 3800 | 19 | 12 | 14 | 103 | 39 |
| *STR10_2b* | 300 | 130 | 500 | 420 | 2700 | 16 | 15 | 10 | 33 | 20 |
| *STR13_a* | 100 | 20 | 250 | 270 | 740 | 15 | 4 | 13 | 54 | 14 |
| *STR13_b* | 140 | 20 | 280 | 340 | 900 | 18 | 5 | 13 | 63 | 16 |
| *STR11_0a* | 8 | 7 | 40 | 6 | 180 | 1 | 1 | 1 | 1 | 2 |
| *STR11_0b* | 8 | 4 | 30 | 4 | 60 | 1 | 1 | 1 | 0.4 | 1 |
| *STR11_1a* | 18 | 10 | 50 | 100 | 330 | 1 | 2 | 1 | 10 | 3 |
| *STR11_1b* | 780 | 90 | 1100 | 1800 | 9000 | 78 | 19 | 40 | 257 | 124 |

**Table 1.** *Cont.*

| | Trace Metal Concentrations | | | | | Enrichment Factors | | | | |
|---|---|---|---|---|---|---|---|---|---|---|
| **Station** | **As** | **Cu** | **Pb** | **Zn** | **Mn** | **As** | **Cu** | **Pb** | **Zn** | **Mn** |
| | $\mu g \cdot g^{-1}$ | $\mu g \cdot g^{-1}$ | $\mu g \cdot g^{-1}$ | $\mu g \cdot g^{-1}$ | $\mu g \cdot g^{-1}$ | | | | | |
| *STR11_2a* | 10 | 9 | 40 | 10 | 100 | 1 | 1 | 1 | 1 | 1 |
| *STR11_2b* | 9 | 4 | 40 | 8 | 140 | 1 | 1 | 1 | 1 | 1 |
| *STR12_b* | 12 | 6 | 30 | 8 | 90 | 1 | 1 | 1 | 1 | 1 |

### *3.2. Ierissos Port*

The map of the total counting rate (TCR), the data interpolation, and the experimental data are depicted in Figure 4 for the beach area of Ierissos.

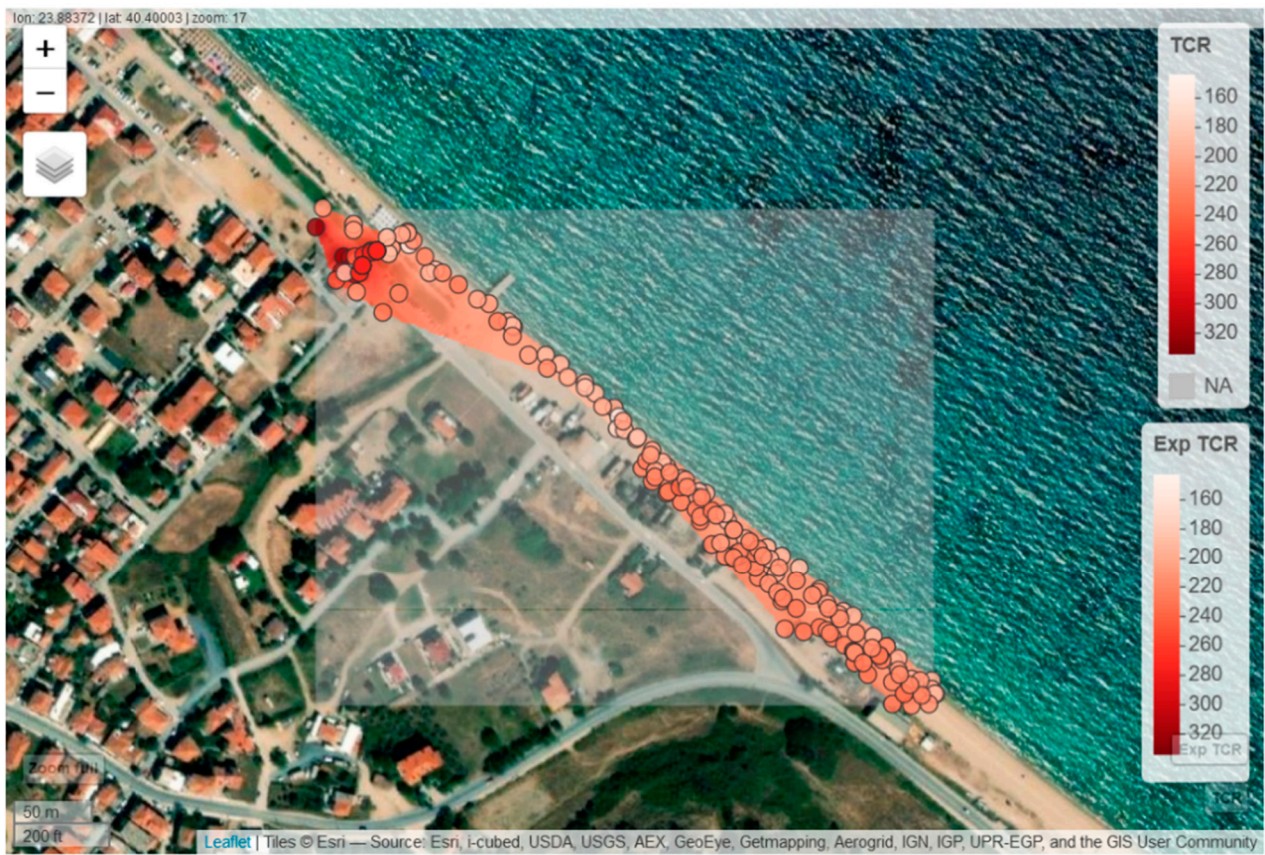

**Figure 4.** The radioactivity distribution of the total counting rate using interpolation methods (for the Ierissos area). The experimental data derived TCR are depicted with circles.

The total counting rate varied from 150 to 300 counts per second in the entire area. A total of 179 spectra were acquired during the experiment, and the live time of the measurement was 3580 s. In the bottom-left area of the spectrum (Figure 5), the spectrum is zoomed in the energy window of the gamma-ray emitters from threshold until 800 keV. In Figure 4, the interpolation data are classified with different colors, while the experimental data are depicted with colored circles. The areas exhibit the highest values in an area close to new construction (see the left part of the experimental data in Figure 4). If these data are omitted, the maxima will then appear in areas where slight streams take place during the winter period. The total counting rate close to the shoreline is significantly reduced compared to the rest area of the beach, since the seawater has no high concentrations of NORMs.

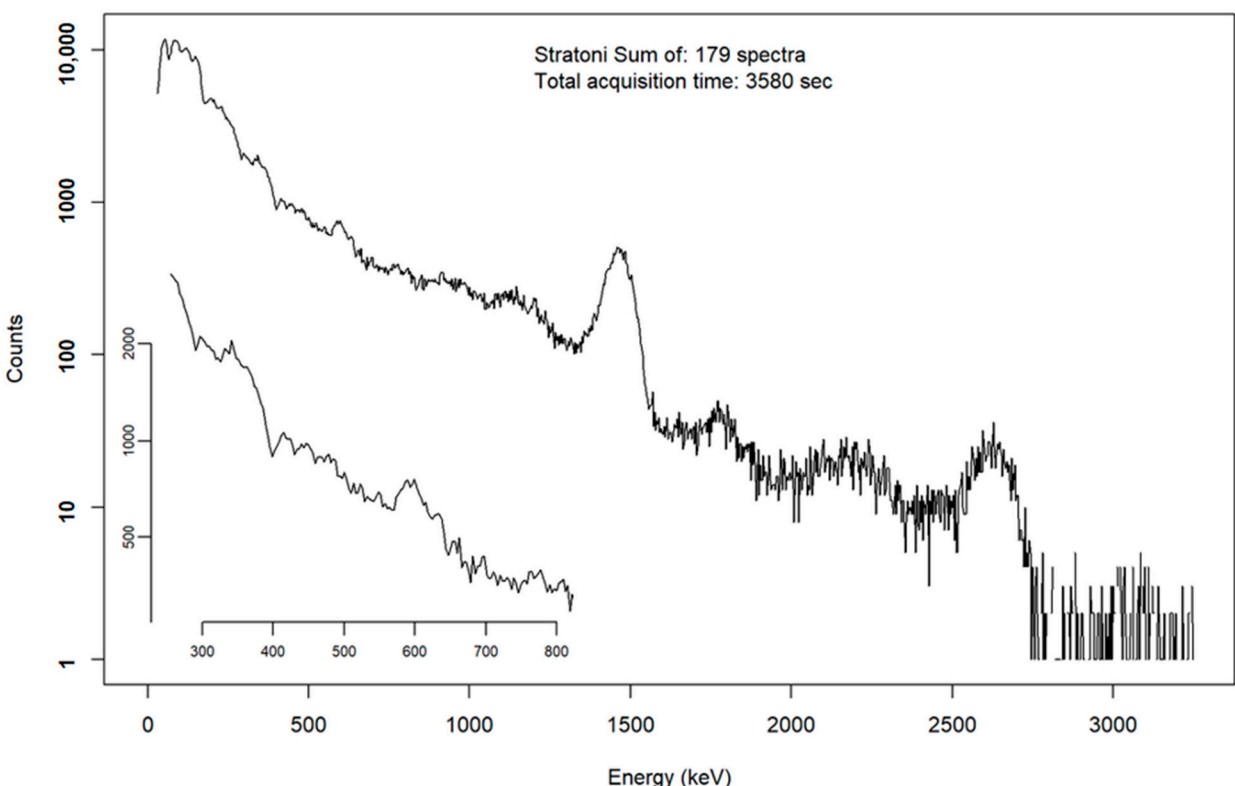

**Figure 5.** The acquired spectrum using the KATERINA II system from the Ierissos beach area. The intense contribution of $^{40}$K is present in the middle of the spectrum (1461 keV). At 1760 and 2614 keV, the contributions of $^{214}$Bi and $^{208}$Tl were used for spectrum analysis, respectively.

The total spectrum as acquired during the entire campaign is depicted in Figure 5, and was produced in an automated manner by the software code. In qualitative terms, the spectrum mainly contains the typical NORMs, such as the progenies of $^{238}$U ($^{214}$Pb and $^{214}$Bi), as well as the progenies of $^{232}$Th ($^{228}$Ac and $^{208}$Tl). As in the previous area of study, the contribution of $^{40}$K (at 1461 keV) was also observed, taking into account that in this photopeak energy area there is also a contribution from several energy peaks of $^{214}$Bi [21,27]. In the Ierissos beach area, the contribution of NORMs (e.g., $^{214}$Bi) in terms of activity concentration is reduced compared to the Stratoni beach area. The spectrum was also analysed for $^{137}$Cs using the SPECTRW software package [27]. Similarly to the previous spectrum for Stratoni beach, sum peaks at high gamma-ray energies are recorded into the spectrum which are not considered for further quantitative analysis.

The laboratory analysis was only performed for one sample in order to determine the activity concentrations, the trace metals, and major elements in the Ierissos beach area. Unfortunately, beach-sea interface sediment samples were not obtained due to the granulometry of the area (grain size >3 mm). The activity concentrations for the $^{238}$U progenies ($^{214}$Pb, $^{214}$Bi, $^{226}$Ra), $^{232}$Th progenies ($^{208}$Tl and $^{228}$Ac), and $^{40}$K were approximately $(9 \pm 1)$ Bq·kg$^{-1}$, $(12 \pm 1)$ Bq·kg$^{-1}$ and $(950 \pm 95)$ Bq·kg$^{-1}$, respectively. The activity concentration of $^{137}$Cs in Ierissos exhibited values similar to those of Stratoni (of $0.8 \pm 0.2$ Bq kg$^{-1}$).

The measured values of As (arsenic), Pb (lead), Zn (zinc), and Cu (copper) were 12 µg g$^{-1}$, 33 µg g$^{-1}$, 8 µg g$^{-1}$, and 6 µg g$^{-1}$, respectively. The measured values of Al (aluminum), Fe (iron), and Mn (manganese) were $56 \times 10^3$·µg g$^{-1}$, $1.3 \times 10^3$·µg g$^{-1}$, and $0.09 \times 10^3$·µg g$^{-1}$, respectively. Among the studied major elements, Al as the most homogeneous element according to the literature [29], can be used as normalizing element. The S (Sulfur) concentration was below the detection limit of the XRF method. The concentrations of Fe (iron), S (sulphur), and Mg (manganese) at Ierissos beach were orders of magnitude lower than those at Stratoni beach, since Ierissos beach is not affected from the mining

activities. The measured data of the studied trace metals in the Ierissos port are provided in Table 1.

## 4. Discussion

### 4.1. Inter-Comparison of Measurement Methods

The activity concentration of in situ measurements (Table 1) was determined by analyzing the spectra depicted in Figures 3 and 5 using SPECTRW software [27]. It should be noted that the contribution of $^{40}$K (at 1461 keV) was observed, considering that this energy range also includes contributions from the various energy peaks of $^{214}$Bi. The low resolution of the detection crystal led to folded spectra, which required unfolding exercises in order to identify the different energy lines [21,27]. The folded spectrum in the range of 580–680 keV resulting from energy peaks at 583 keV (emitted by $^{208}$Tl), 609 keV (emitted by $^{214}$Bi), and 661 keV (emitted by $^{137}$Cs), was de-folded using the appropriate routine of the software [27]. Table 2 presents the activity concentration of observed radionuclides, obtained through laboratory-based methodologies, for the beaches of Stratoni and Ierissos. Additionally, the activity concentration of samples from the main rivers (Kokkinolakkos and Kalatzi Lakkos) of the Ierissos Gulf is provided. The in situ data were spatially averaged, whereas the laboratory analysis involved point sampling.

**Table 2.** The activity concentration in Bq·kg$^{-1}$ using the in situ and lab-based gamma ray spectrometry method in the Stratoni and Ierissos beach sand areas.

| | In Situ | | Laboratory | | |
|---|---|---|---|---|---|
| **Isotope** | **Stratoni** | **Ierissos** | **Stratoni** | **Ierissos** | **Rivers** |
| $^{40}$K | 425 ± 34 | 417 ± 33 | 580 ± 60 | 950 ± 95 | (320–1100) ± 100 |
| $^{214}$Bi | 33 ± 4 | 12.3 ± 1.5 | 50 ± 4 | 9.3 ± 0.9 | (8–60) ± 3 |
| $^{208}$Tl | 19 ± 2 | 10.0 ± 1.2 | 25 ± 2 | 12.2 ± 1.1 | (11–30) ± 4 |
| $^{137}$Cs | 6.9 ± 1.1 | 3.8 ± 0.5 | 0.8 ± 0.2 | 0.8 ± 0.2 | (0.7–20) ± 0.2 |

The lab-based gamma ray technique yielded activity values that were 20% to 50% higher than the in situ method for both $^{214}$Bi and $^{208}$Tl in the Stratoni and Ierissos beach areas. Regarding $^{40}$K, the two methodologies showed agreement (with a 36% divergence) in Stratoni beach, but they significantly differed with regard to Ierissos beach, where the lab-based method provided $^{40}$K data that was twice as high compared to the in-situ analysis. As for $^{137}$Cs, the two methods produced distinct results, indicating that the in situ activity concentrations in the Stratoni and Ierissos beach areas were nine and five times higher than the lab-based measurements, respectively. This difference can be attributed to the low activity concentrations (<1 Bq kg$^{-1}$) of $^{137}$Cs and the challenges associated with deconvoluting the 580-680 keV energy range, which involves contributions from three main radionuclides ($^{214}$Bi, $^{208}$Tl, and $^{137}$Cs), as mentioned earlier. The activity concentrations of NORMs and $^{137}$Cs obtained from the rivers indicate areas where mapping should be conducted, as the activity values exhibit a significant variation, with higher values suggesting potential radionuclide hotspots. The ongoing environmental remediation efforts in the Stratoni study further complicate the consistency of the results.

It is important to note that any comparison of the two methods should not be based on the average measured values of the activity concentration of radionuclides. The in situ method provides a spatial average value across the entire beach area, while the lab-based method involves random (or grid) point sampling and subsequent measurements using high-resolution gamma-ray spectrometry. The in situ method offers advantages such as rapid site characterization, cost-effectiveness, and the ability to identify potential areas of high activity concentration or hotspots resulting from fallout in specific beach locations. On the other hand, the lab-based method offers greater measurement accuracy, but if the

sampling point is distant from the fallout area, it may not be identified, leading to an erroneous site characterization.

*4.2. Radiological Assessment*

The data from both the in situ and lab-based methods in this study were utilized for calculating the dose rates resulting from the activity concentration of natural radionuclides in the beach sand of the two areas. At Stratoni beach, the dose rate in the sand area was determined to be 45 nGy h$^{-1}$. This value is enhanced by approximately 30% when compared to assessments conducted in the open waters of the Adriatic Sea near magmatic regions [30], as well as with assessments (30 to 32 nGy h$^{-1}$) reported in the Gulf of Aqaba. However, the radiological values of Stratoni beach are comparable to industrial areas in the Red Sea [31], but significantly lower than phosphate ports (263 nGy h$^{-1}$). The slightly higher dose rate values at Stratoni beach may be attributed to the discharge of washout materials enriched with past mining activities where material entered into the central part of the beach sand through a stream, in addition to the discharge of mining by-products and wastes. In the case of Ierissos beach, the dose rate in the sand area was determined to be 29 nGy h$^{-1}$, which is similar to the assessments conducted in other areas, such as near magmatic regions in the Adriatic Sea [29]. Similar values (30 to 32 nGy h$^{-1}$) have been reported in the Gulf of Aqaba in the Red Sea [31] in locations that are far from industrial areas.

Comparing the dose rates determined by the lab-based data with the in situ method, the lab-based method yielded values that were 40% and 77% higher than the in situ method for the Stratoni (45 nGy h$^{-1}$) and Ierissos (29 nGy h$^{-1}$) beaches, respectively. This discrepancy may be attributed to the fact that the in situ method effectively takes into account the variability of the activity concentration by covering a larger spatial area, whereas the lab-based method requires a denser sampling of the beach area. Furthermore, the dose rates in the river areas ranged from 30 to 102 nGy h$^{-1}$, indicating promising areas for rapid and safe mapping. Nevertheless, the dose rates obtained for the Stratoni and Ierissos beaches, as well as the rivers, were within the dose rate ranges (30–109 nGy h$^{-1}$) reported in Greece for external exposure rates due to terrestrial gamma radiation [32].

*4.3. Enrichment Factors*

The enrichment factors (EF) for the trace metals (As, Cu, Pb, Zn) and Mn are provided in Table 1. The EF values of the aforementioned trace metals exhibited a wide range, with values varying from 1 to 750, 1 to 109, 1 to 112, 1 to 987, and 1 to 482, respectively. According to the literature [33], EF values above 10, 25, and 50 indicate the severe, very severe, and extremely severe enrichment of the studied element, respectively. Based on these classifications, Stratoni beach is characterized as extremely severely enriched in As, Zn, and Mn; very severely enriched in Pb; and severely enriched in Cu. It is worth noting that a yellow-gold colored sample collected from the beach sand (within the first 5 cm) in the Stratoni area was found to be very severely enriched in all of the studied trace metals. This suggests that the layers below the surface in the remaining beach area may contain even higher metal concentrations. It is important to note that only surface sediments, such as beach sand, were collected from the studied areas. The river samples also displayed significant enrichment in As, Cu, Pb, Zn, and Mn, with some sampling points reaching extremely severe levels of enrichment.

Regarding remediation strategies, various methods such as soil amendment (using adsorption and precipitation) and phytoremediation techniques have been applied for environmental cleanup and restoration in similar areas. Additionally, traditional methods such as excavation and ex situ treatment can be considered, provided that international protocols are followed. To evaluate the effectiveness of remediation activities, detailed maps depicting the concentrations of radionuclides and metals using the in situ method are crucial to fulfil spatial requirements. These maps will assist national authorities in

characterizing and assessing the Stratoni area following the remediation process in terms of metal enrichment and radiological doses.

## 5. Summary

In this work, a rapid and safe human mapping characterization was performed in a pre-remediated area (Stratoni) and an area unaffected by human activities (the Ierissos region). The metal mine located near Stratoni was greatly affected over the years not only for the terrestrial part, but also with regard to the marine compartment of the area. The impact of metal was proven by the high enrichment factor values and by the non–homogenous NORM spatial distribution. The rapid mapping revealed the areas with the highest TCR, and thus the in situ activity concentrations provided a more detailed, low cost and representative NORM spatial distribution map for the area of study that could not be solely accomplished by sampling and lab measurements. Despite the low NORM discrepancies found in the studied areas in terms of activity concentrations, this methodology has proven to be a good tool for other likely affected areas (such as neighboring rivers) or sites with high NORM concentrations (e.g., coal mines and phosphogypsum factories).

**Author Contributions:** Data curation, C.T., D.L.P. and F.K.P.; Formal analysis, D.L.P. and F.K.P. Investigation, A.L., C.T., R.A., J.C., M.F.H., C.H., L.M., L.K., M.V., F.K.P. and D.L.P.; Methodology, C.T., S.A. and D.L.P.; Software, S.A.; Supervision, A.L. and C.T.; Validation, C.T. and F.K.P.; Visualization, C.T., R.A., J.C., M.F.H., C.H., L.M., L.K., S.A. and D.L.P.; Writing—original draft, C.T., A.L., D.L.P. and F.K.P.; Writing—review & editing, C.T., D.L.P., F.K.P., R.A., J.C., M.F.H., C.H., L.M. and L.K. All authors have read and agreed to the published version of the manuscript.

**Funding:** This research was not funded by an authority.

**Institutional Review Board Statement:** The study was conducted in the frame of training activity in a mining area approved for submission by the Institutional Director of the Institute of Oceanography of HCMR.

**Informed Consent Statement:** Informed consent was obtained from all subjects involved in the study.

**Data Availability Statement:** The data presented in this study are available upon request to the corresponding author.

**Acknowledgments:** The authors would like to acknowledge IAEA for supporting this activity through TC project RER1020, as well as for supporting the sediments dynamics evaluation studies through the Coordinated Research Project F22074, "Development of Radiometric Methods and modelling for measurement of sediment transport in coastal systems and rivers". The participating students of the City College of New York (CCNY), along with their faculty representative, Angelo Lampousis, gratefully acknowledge Dean Juan Carlos Mercado of the Division of Interdisciplinary Studies at the Center for Worker Education of the City University of New York (CUNY), as well as Ninive Gomez of the Office of Study Abroad and International Programs at CCNY. They also gratefully acknowledge the American Farm School of Thessaloniki in Greece for the significant logistical support with regard to equipment repair and transportation.

**Conflicts of Interest:** The authors declare that they have no conflicts of interest.

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
