# Peer review of "In Situ Radioactivity Maps and Trace Metal Concentrations in Beach Sands of a Mining Coastal Area at North Aegean, Greece"

_jmse, doi:10.3390/jmse11061207_

Round 1

Reviewer 1 Report

Introduction

I noticed too many statements without citations. I presumed most of these statements did not originate from you. For example the statements in lines 37-41 were not cited; please provide citation(s). Kindly revisit and cite appropriately.

I would also advise the authors to spell out the chemical symbols of the trace metals and other elements presented in this write up so that the lay readers will be carried along all through.   

The arguments were not flowing, and I will advise you revisit. Some of the statements were just floating with no direction.

Please see the attached documents for detailed comments.

Materials and methods    

 The section was well written except in some areas. Please see the attached documents for detailed comments.

Results

The authors appear to be discussing their results instead of interpreting here, please revisit and interpret results only. You can find other comments on the attached documents.

Discussion  

The discussion should be revisited by the authors and they were wavering between results interpretation and discussion. At some point they were also presenting their methods here.

Please see the attached documents for detailed comments.

Moderate English editing will suffice.

Reviewer 2 Report

Manuscript ID: jmse-2375605-peer-review-v1

Title: In-situ radioactivity maps and trace metal concentrations in beach sands of a mining coastal area at North Aegean, Greece

Comments:

Line 19-20: “Lab-based measurements collected prior to the in-situ data collections for the determination of radionuclide and metal concentrations.” Incomplete sentence. Please check it.

Line 24: “an enrichment factor”. Detailed index name should be provided.

Line 26-27: “We identify as a source of uncertainty the fact that one of the two sites, Stratoni, was undergoing active remediation concurrently with our in-situ data collection activities.” This sentence is not clear for me. Please rewrite it.

Line54-55 : It should be “such as climate change studies [9] and marine pollution [7], [8]”. And, ( should be deleted.

Line 56-60: Please provide some references here.

Line 68: A specific research gap should be given here to introduce the aim of this study, rather than just the background understanding of the 137Cs. The authors should summary it at the end of this paragraph.

Line 99-102: Incomplete sentences. Please check them.

Line 114: Whether there are some corresponding references?

Line 319: Please check the superscript.

Line 329: Much more detailed discussions about the difference of the lab-base and in-situ method results should be given. Moreover, the potential methods to avoid the significant difference should be proposed.

Line 343, 351: “(Petrinec et al., 2010)”...... Inconsistent reference format. Please check it.

Line 356-357: This sentence is not clear for me. Please rewrite it.

Line 364: Please provide the relevant results in forms of figure or table.

Line 376: Much more discussions about the enrichment factor assessment of trace metals and the comparison with other studies, for example, other mining-contaminated coastal areas, should be provided here, in order to reflect the unique pollution characteristics in the mining coastal area at North Aegean, Greece (this study) and to provide the potential remediation strategies.

English should be polished. There are numerous errors in the current manuscript with many grammatical mistakes and incorrect presentations. These errors make the article difficult to read and understand.

Reviewer 3 Report

Dear Auhors

The study has many serious flaws:

1) A coherent sampling program has not been established.
2) The laboratory analysis procedures followed for the determination of the isotopes studied have not been described.
3) The mapping method has not been described and the software used has not been indicated.
4) No control or reference zones have been established.

5) Image 1 has very poor resolution
6) There are errors in the way bibliographic citations are indicated (e.g. Lines, 49, 55, 95...).

Round 2

Reviewer 3 Report

Dear Authors

Thanks for your improved manuscript, it can be accepted in present form